# Multivariate Techniques for Monitoring Susceptible, Exposed, Infected, Recovered, Death, and Vaccination Model Parameters for the COVID-19 Pandemic for Qatar

**DOI:** 10.3390/ijerph21121580

**Published:** 2024-11-27

**Authors:** Abdel-Salam G. Abdel-Salam, Edward L. Boone, Ryad Ghanam

**Affiliations:** 1Department of Mathematics and Statistics, College of Arts and Sciences, Qatar University, Doha P.O. Box 2713, Qatar; 2Department of Statistical Sciences and Operations Research, College of Humanities and Sciences, VCU, Richmond, VA 23284, USA; elboone@vcu.edu; 3Department of Liberal Arts and Science, VCU in Qatar, Doha P.O. Box 1129, Qatar; raghanam@vcu.edu

**Keywords:** MEWMA, MCUSUM, SEIRDV, epidemiology, augmented particle MCMC, profile monitoring, COVID-19

## Abstract

The COVID-19 pandemic has highlighted the crucial role of health sector decision-makers in establishing and evaluating effective treatment and prevention policies. To inform sound decisions, it is essential to simultaneously monitor multiple pandemic characteristics, including transmission rates, infection rates, recovery rates (which indicate treatment efficacy), and fatality rates. This study introduces an innovative application of existing methodologies: the Multivariate Exponentially Weighted Moving Average (MEWMA) and Multivariate Cumulative Sum (MCUSUM) control charts (CCs), used for monitoring the parameters of the Susceptible, Exposed, Infected, Recovered, Death, and Vaccination (SEIRDV) model. The methodology is applied to COVID-19 data from the State of Qatar, offering new insights into the pandemic’s dynamics. By monitoring changes in the model parameters, this study aims to assess the effectiveness of interventions and track the impact of emerging variants. The results underscore the practical utility of these methodologies for decision-making during similar pandemics. Additionally, this study employs an augmented particle Markov chain Monte Carlo scheme that enables real-time monitoring of SEIRDV model parameters, offering improved estimation accuracy and robustness compared to traditional approaches. The results demonstrate that MEWMA and MCUSUM charts are effective tools for monitoring SEIRDV model parameters and can support decision-making in any similar pandemic.

## 1. Introduction

Coronavirus disease 2019 (COVID-19) was a devastating pandemic that hit every part of the world [1,2]. Because of the rapid rate at which it was spreading, appropriate safety measures had to be taken to keep it under control. Because there was currently only a limited potential cure and therapy for the target, the establishment of those safeguards was unavoidable. These restrictions can be classified as social separation, closure of enterprises or schools, and prohibitions on travel [3,4]. The COVID-19 virus is classified under the same family as Nidovirales, which are viruses that employ a nested set of mRNAs for replication. Additionally, it is categorized within the subfamily of alpha, beta, gamma, and delta Co-Vis viruses. The virus responsible for the onset of COVID-19 is classified within the Betacoronavirus 2B lineage and exhibits a tight genetic relationship with SARS species. The virus in question is considered new due to the limited binding affinity of monoclonal antibodies towards SARS-CoV-2. Viral RNA replication occurs through the binding and subsequent re-attachment of RNA polymerase to various sites [5,6].

COVID-19 began in December of 2019 when doctors in Wuhan, China, reported a cluster of unusual symptoms [1,2]. As a result, the COVID-19 virus quickly reached other regions of the world. The United States reported the first case of COVID-19 on 20 January 2020; on 31 January 2020, Italy reported its first verified case. As the number of reported cases of COVID-19 increased around the world, governments were quick to step in and regulate the banking and medical industries. The United States put its initial travel restrictions on Chinese citizens in late January 2020. A few weeks later, many countries in Europe and the United Kingdom instituted their own travel bans. On 11 March 2020, with more than 100,000 cases worldwide, the World Health Organization (WHO) designated COVID-19 a pandemic. On 26 January 2021, the global tally of confirmed cases of COVID-19 surpassed 100 million, with over 2.15 million deaths. The global fatality rate from this virus is 3.4%, making it more dangerous than the flu. COVID-19 has a higher mortality rate in the elderly, especially those with other health problems. Those with just minor symptoms typically feel better within 3–7 days, while those with pneumonia or other severe disorders may need weeks to fully recover [7]. To give just one example, as of 26 January 2021, the percentage of patients who made a full recovery in China and India is 95%. The estimated global COVID-19 recovery rate is 97%.

The COVID-19 pandemic has significantly impacted public health and the global economy. Governments worldwide have implemented various measures to control the spread of the virus, such as lock-downs, testing and tracing, and vaccination programs. Mathematical models have played a crucial role in understanding the spread of the virus and predicting its impact on public health. The Susceptible, Exposed, Infected, Recovered, Death, and Vaccinated (SEIRDV) model is one of the most widely used mathematical models for analyzing and predicting infectious diseases. The SEIRDV model divides the population into different compartments based on their disease status and accounts for various parameters such as transmission, recovery, and vaccination rates. In this study, we propose a statistical monitoring approach using the Multivariate Exponentially Weighted Moving Average (MEWMA) and Multivariate Cumulative Sum (MCUSUM) Control Charts (CCs) to monitor the parameters of the SEIRDV model for the COVID-19 pandemic in the State of Qatar.

As opposed to monitoring the process, the literature focuses primarily on model estimation and forecasting the dynamic nature of the COVID-19 pandemic. The SEIRDV model is a common compartmental model used to model the spread of disease through a population, and variants of this model have been utilized to model the COVID-19 pandemic, for example, [8] for Italy, [9] for India, and [10] for the State of Qatar. For more on disease modeling in general, see [11,12,13,14].

Other modeling approaches include a time-series model to analyze the pandemic’s outbreak and a time-varying Bayesian semi-parametric model to examine the pandemic’s short-term projections [15,16,17,18].

Several authors have examined the dynamic pattern of COVID-19-associated fatalities over time [3,19]. In China [20] and India [9,21], the impact of government interventions, such as a lock-downs, has been examined. The COVID-19 outbreak in Ukraine was monitored using mathematical modeling approaches [22]. Zhang et al. [23] investigated enhancements to traditional epidemiological models to better capture the virus’s complex spread dynamics. Their research highlights the importance of integrating real-time data into model parameters, which supports more accurate predictions and enables more effective public health interventions. By applying advanced statistical techniques to refine the models’ sensitivity to changes in infection rates and public health interventions, they demonstrated the critical role of dynamic modeling in pandemic management. Nevertheless, the parameter estimates are not obtained directly from the data, and there is no explicit tracking system provided. As the vaccine is being disseminated to the public, it is crucial for decision-makers to closely monitor the pandemic. This will enable them to assess the progress of the pandemic and detect any possible disruptions to the system. The monitoring approach must possess the capability to promptly respond to any substantial alterations in the system. However, numerous studies have employed SEIRDV models for pandemic analysis, and multivariate control charts like MEWMA and MCUSUM have proven valuable in process control. There are no studies that have combined these methodologies to address pandemic monitoring. Unlike prior research focusing on single methodologies or different pandemic contexts, our study offers a comprehensive approach, using MEWMA and MCUSUM charts in conjunction to provide a holistic view of pandemic dynamics.

In response to this need, this study introduces a general approach utilizing Multivariate Exponentially Weighted Moving Average (MEWMA) and Multivariate Cumulative Sum (MCUSUM) control charts. These control charts show how the dynamic parameters of the SEIRDV model change over time. SEIRDV is a complete epidemiological model that shows how diseases spread through different stages. Our methodology focuses on the general applicability of these control charts in epidemiological monitoring, which we later contextualize with specific application to COVID-19 data from the State of Qatar. By initially outlining the general monitoring strategy, in this study we aim to provide a foundation that can be adapted to various epidemiological scenarios.

The present study acquired data pertaining to COVID-19 cases and fatalities in the State of Qatar from the World Health Organization (WHO). The data were subsequently subjected to analysis using the SEIRDV model. The maximum likelihood estimation method was employed to calibrate the SEIRDV model, and subsequently the model parameters were calculated for each day of the study period. The MEWMA and MCUSUM control charts were then employed to monitor the temporal variations in the model parameters.

The objective of this study is to devise a technique for monitoring a pandemic by utilizing the mathematical model, SEIRDV, which can be swiftly revised in response to emerging data and can detect any anomalous alterations in the parameters of the mathematical model. There do not appear to be any approaches in the literature that meet this objective. One of the most challenging aspects of this subject pertains to the automation of parameter modifications and their subsequent conversion into a monitorable format. In this study, a Bayesian approach is employed to estimate parameters using a sampling algorithm. The Bayesian approach is employed to dynamically estimate the time-varying parameters of the SEIRDV model using real-time data. The technique involves constructing posterior distributions for the parameters at each time step, which are then updated sequentially as new data become available. This algorithm allows for quick updates, prevents the elimination of particles, and produces samples that can be monitored using the Multivariate Exponentially Weighted Moving Average (MEWMA) and Multivariate Cumulative Sum (MCUSUM) control charts. In addition, this study proposes an innovative application of MEWMA and MCUSUM control charts to the SEIRDV model for tracking pandemic parameters. Although these methodologies are well-established individually, their combination and specific application to COVID-19 provide novel insights. Our aim is to empower decision-makers with enhanced tools for real-time pandemic monitoring and control. Hence, by employing both MEWMA and MCUSUM, our framework balances sensitivity (to detect gradual changes) and robustness (to identify abrupt changes), providing a comprehensive monitoring system. This dual approach enhances decision-making compared to methods that rely solely on univariate or single-purpose techniques.

This work is structured as follows: Section 2 introduces the SEIRDV model, the mean model, and the likelihood methods used. In Section 3, we present the Multivariate Exponentially Weighted Moving Average (MEWMA) and Multivariate Cumulative Sum (MCUSUM) techniques, which are designed to track multiple variables simultaneously. The model parameters are updated at each time step through the Markov Chain Monte Carlo (MCMC) sampling algorithm with particle augmentation, as detailed in Section 4. Section 5 demonstrates how these methods are applied using real-life data from the State of Qatar to detect significant shifts in the pandemic. Finally, Section 6 discusses the methodology, offers implementation tips, and identifies areas for potential improvement.

## 2. The SEIRDV Model for Predicting Outcomes with Vaccination

The Susceptible, Exposed, Infected, Recovered, Death, Vaccinated (SEIRDV) model extends traditional compartmental models by explicitly incorporating vaccination status, which is critical in pandemic monitoring. Each compartment in the SEIRDV model represents a specific stage in the pandemic’s progression, influenced by disease transmission, intervention measures, and vaccination efforts, where **Susceptible (S)** represents individuals who have not been exposed to the virus and remain at risk of infection, **Exposed (E)** represents individuals who have been exposed to the virus but are not yet infectious, **Infected (I)** represents individuals who are infectious and can transmit the virus to susceptible individuals, **Recovered (R)** represents individuals who have recovered from the infection and are considered immune, **Death (D)** represents individuals who have died from the infection, and **Vaccinated (V)** represents individuals who have received a vaccine, which significantly impacts their susceptibility to infection.

In this study, we utilize a comprehensive compartmental model (SEIRDV) to track COVID-19 dynamics, represented by ordinary differential equations. Let *t* denote a time index representing the duration in days since the initial documented instance of COVID-19 among the targeted population. Let S(t) represent the number of individual subjects who are susceptible at time *t*; E(t) represent the number of individuals who are exposed at time *t*; I(t) represent the number of individuals who are infected (symptomatic) at time *t*; R(t) represent the total number of individuals who have recovered at time *t*, where R(t) is split into $R_{E}\left(t\right)$ for those recovered from exposed and $R_{I}\left(t\right)$ for those recovered from infected; D(t) represent the total number of individuals who have died at time *t*; and V(t) represent the total number of individuals who have been vaccinated at time *t*. The following system of ordinary differential equations can be used to model this:(1)$$\begin{array}{ccc} \frac{dS}{dt} & = & -\alpha S(t)E(t)-\rho S(t) \end{array}$$(2)$$\begin{array}{ccc} \frac{dE}{dt} & = & \alpha S(t)E(t)-(\beta +\gamma +\rho )E(t) \end{array}$$(3)$$\begin{array}{ccc} \frac{dI}{dt} & = & \beta E(t)-(\gamma +\eta )I(t) \end{array}$$(4)$$\begin{array}{ccc} \frac{dR_{E}}{dt} & = & \gamma E\left(t\right)-\rho R_{E}\left(t\right) \end{array}$$(5)$$\begin{array}{ccc} \frac{dR_{I}}{dt} & = & \gamma I(t) \end{array}$$(6)$$\begin{array}{ccc} \frac{dD}{dt} & = & \eta I(t) \end{array}$$(7)$$\begin{array}{ccc} \frac{dV}{dt} & = & \rho S\left(t\right)+\rho E\left(t\right)+\rho R_{E}\left(t\right), \end{array}$$
under the following restrictions $\;S\left(t\right)\geq 0,\;E\left(t\right)\geq 0,\;I\left(t\right)\geq 0,\;R_{E}\left(t\right)\geq 0,\;R_{I}\left(t\right)\geq 0,$ D(t)≥0, and V(t)≥0. The following explanations are given for the parameters: α denotes the daily transmission rate from susceptible to exposed, calculated as per day per individual squared, reflecting the probability of contact and transmission, and β represents the daily rate at which exposed individuals become infected (symptomatic). γ indicates the recovery rate, applied equally to symptomatic and asymptomatic individuals, assuming uniform recovery dynamics, and η represents the daily rate of death for those infected. In addition, the parameter ρ plays a critical role in representing the efficacy and coverage of vaccination within the model, providing a quantitative measure of how vaccination rates affect the overall disease spread by directly lowering the rates of transmission and exposure in the population. It is important to note that this model formulation makes the following assumptions. The model does not account for immigration, emigration, natural mortality, or births since these variables have a minor impact on the time period. After being classified as part of the infected group, an individual is required to remain in quarantine and avoid contact with the susceptible population. Those who first become infected are accommodated in the recovered and deaths sections. Recoveries occur at the same rate, γ, for both symptomatic and asymptomatic individuals exposed to the virus. The assumptions required for the example given in Section 5 are met by the SEIRDV model that is provided here, which is a modified version of the one provided in [10]. This model, while not unique in its basic structure, features a distinct approach to estimation and monitoring. Before delving into the specifics of the model equations, it is crucial to define the Poisson λ parameters, which represent the expected number of individuals in each compartment per unit time. These parameters are essential for understanding the dynamics of disease spread within the population. The Poisson λ parameters are defined as follows:$\lambda_{s}\left(t\right)$: Expected number of susceptible individuals at time *t*.$\lambda_{E}\left(t\right)$: Expected number of exposed individuals at time *t*.$\lambda_{I}\left(t\right)$: Expected number of infected individuals at time *t*.$\lambda_{R}\left(t\right)$: Expected number of recovered individuals at time *t*.$\lambda_{D}\left(t\right)$: Expected number of deceased individuals at time *t*.
(8)$$\begin{array}{lll} \frac{d\lambda_{S}\left(t\right)}{dt} & = & -\alpha \left(t\right)\lambda_{S}\left(t\right)\lambda_{E}\left(t\right)\\ \frac{d\lambda_{E}\left(t\right)}{dt} & = & \alpha \left(t\right)\lambda_{S}\left(t\right)\lambda_{E}\left(t\right)-\beta \left(t\right)\lambda_{E}\left(t\right)-\gamma \left(t\right)\lambda_{E}\left(t\right)\\ \frac{d\lambda_{I}\left(t\right)}{dt} & = & \beta \left(t\right)\lambda_{E}\left(t\right)-\gamma \left(t\right)\lambda_{I}\left(t\right)-\eta \left(t\right)\lambda_{I}\\ \frac{d\lambda_{R}\left(t\right)}{dt} & = & \gamma \left(t\right)\lambda_{I}\left(t\right)\\ \frac{d\lambda_{D}\left(t\right)}{dt} & = & \eta \left(t\right)\lambda_{I}\left(t\right), \end{array}$$

At any given time point, *t*, it is necessary for the parameters to possess a previous distribution. In this study, the previous distribution specification remains consistent throughout all time intervals, *t*. However, it is not essential to adhere to this requirement if there is special information that has to be included at a particular point in time. Given the use of a Bayesian technique, the probability is explicitly defined as:(9)$$\begin{array}{lll} I(t) & ∼ & Poisson\left(\lambda_{I}\left(t\right)\right)\\ R(t) & ∼ & Poisson\left(\lambda_{R}\left(t\right)\right)\\ D(t) & ∼ & Poisson\left(\lambda_{D}\left(t\right)\right)\\ V(t) & ∼ & Poisson\left(\lambda_{V}\left(t\right)\right). \end{array}$$

It should be noted that the variables S(t) and E(t) are not included in the probability due to their latent nature, meaning they are not directly observed. Observing the number of exposed subjects is challenging due to the possibility of a large number of individuals in the community being exposed to and carrying the virus. However, these individuals may not show any symptoms and hence cannot be easily detected. Due to the visual indistinguishability between asymptomatic exposed individuals and those who are vulnerable, the direct observation of susceptible individuals poses a challenge. The data in question were not collected by the government, resulting in its unobservability or latent nature. The appropriate probability distribution for the variables (S(t),E(t),I(t),R(t), and D(t)) should be a multinomial. Nevertheless, the application of the multinomial technique becomes hard when dealing with two latent states, one of which is the larger state. Consequently, this study uses Poisson probability as an approximation.

At each time step, the technique modifies the posterior distribution of the parameters. The posterior distribution for the parameters at each time step is generated, enabling further monitoring using multivariate monitoring techniques such as MEWMA charts, MCUSUM charts, or other similar methods. One can see that incorporating vaccination status into pandemic modeling presents unique challenges. The varying efficacy of different vaccines introduces complexity to the model, influencing transition probabilities between compartments. Additionally, vaccine hesitancy, or reluctance to vaccinate in certain populations, complicates model parameters and necessitates additional data for accurate predictions. The pace of vaccine roll-out further impacts the model’s dynamics, particularly in transitioning individuals from susceptible to vaccinated compartments. These challenges significantly affect pandemic monitoring. Changes in the vaccinated compartment can signal the emergence of new variants or waning immunity, necessitating intervention adjustments. Monitoring the impact of vaccination on disease transmission helps policymakers assess the effectiveness of vaccination campaigns and adapt strategies accordingly. Furthermore, understanding transitions from infection to recovery or death aids in predicting healthcare demand and optimizing resource allocation.

The SEIRDV model describes disease progression through the susceptible, exposed, infected, recovered, death, and vaccination compartments. MEWMA and MCUSUM control charts effectively monitor SEIRDV model parameter variations to detect changes in pandemic progression.

## 3. Multivariate Profile Monitoring Techniques

The approach proposed for estimating the parameters of the SEIRDV model over time demonstrates a high degree of responsiveness to variations in the system. Therefore, it is possible to monitor these characteristics in order to detect any alterations that may be evident in the data. To effectively monitor SEIRDV model parameters, this study utilizes Multivariate Exponentially Weighted Moving Average (MEWMA) and Multivariate Cumulative Sum (MCUSUM) control charts. MEWMA charts monitor shifts in multivariate data over time, emphasizing recent data to provide cumulative monitoring. MEWMA charts excel in their sensitivity to small changes, effectively detecting subtle shifts in pandemic progression. It also accounts for correlations between various pandemic parameters, providing a holistic view of the model dynamics, and the exponential weighting smooths out data noise, highlighting significant trends. On the other hand, MCUSUM charts accumulate deviations, making them more sensitive to sudden shifts. They are particularly effective in detecting abrupt and significant changes, which are crucial for rapid response. Their distinct detection separates substantial deviations from normal variations, making them ideal for detecting unexpected changes. By employing both MEWMA and MCUSUM charts, this approach leverages MEWMA’s and MCUSUM’s sensitivity to small to moderate prospective changes, creating a comprehensive monitoring system. MEWMA can alert decision-makers to gradual shifts early on, while MCUSUM charts provides a follow-up to confirm significant deviations, ensuring a more accurate response. The combination offers an early warning system capable of identifying subtle and significant changes in the pandemic’s trajectory. It also provides actionable insights for healthcare planning and policy adjustments, aiding in prompt decision-making.

### 3.1. The MEWMA Control Chart

The MEWMA control chart is utilized to monitor multivariate processes over time. While Roberts introduced the concept of the Exponentially Weighted Moving Average (EWMA) chart in 1959 [24], focusing on univariate data, the extension to multivariate data, the multivariate EWMA control chart, was developed by Lowry et al. (1992) [25]. This development allowed the principles of the univariate EWMA to be applied to multivariate processes, enhancing its utility in monitoring complex systems.

The MEWMA control chart has several advantages for monitoring the SEIRDV model parameters during the COVID-19 pandemic. Firstly, the MEWMA control chart is able to account for the correlation between the different variables in the model, which is important for understanding the complex relationships between the model parameters. Secondly, the MEWMA control chart is able to detect small and gradual changes in the model parameters, which is useful for monitoring the effectiveness of interventions and the impact of new variants. Finally, the MEWMA control chart is a well-established statistical process control tool, which means that it is based on a solid foundation of statistical theory and practice. It is designed to be sensitive to small shifts in the mean vector while remaining robust to larger shifts.

In our study, the daily transmission rate, α(t), is modeled as a time-varying parameter to reflect real-time changes in transmission dynamics driven by external factors such as public health interventions, behavior changes, and new variants. We initially apply a backward lag differencing technique to the time series data of each parameter with a lag of one. This produces the differences $\Delta \alpha {\left(t\right)}_{i}=\alpha {\left(t\right)}_{i}-\alpha {\left(t-1\right)}_{i}$, $\Delta \gamma {\left(t\right)}_{i}=\gamma {\left(t\right)}_{i}-\gamma {\left(t-1\right)}_{i}$, $\Delta \beta {\left(t\right)}_{i}=\beta {\left(t\right)}_{i}-\beta {\left(t-1\right)}_{i}$, and $\Delta \eta {\left(t\right)}_{i}=\eta {\left(t\right)}_{i}-\eta {\left(t-1\right)}_{i}$. These differences were compiled into a vector, ${\left(\Delta \alpha {\left(t\right)}_{i},\Delta \gamma {\left(t\right)}_{i},\Delta \beta {\left(t\right)}_{i},\;\Delta \eta {\left(t\right)}_{i}\right)}^{T}$, which is scrutinized for notable deviations from zero, indicative of significant alterations in the parameter values. The aggregate changes in the parameters, characterized by their mean, $\bar{\theta }\left(t\right)$, are defined as:$$\Delta \bar{\theta }\left(t\right)=\frac{1}{n_{p}}1_{n_{p}}^{T}\left(\Delta \alpha (t),\Delta \gamma (t),\Delta \beta (t),\Delta \eta (t)\right)$$
where θ(t) represents this parameter vector, which encompasses the time-varying model parameters (such as α(t),γ(t),β(t),η(t)) that define transition rates within the SEIRDV framework, with their variance described by:$$Cov\left(\Delta \theta \left(t\right)\right)=\frac{1}{n_{p}-1}\left(\begin{array}{c} \Delta \alpha {\left(t\right)}^{T}\\ \Delta \gamma {\left(t\right)}^{T}\\ \Delta \beta {\left(t\right)}^{T}\\ \Delta \eta {\left(t\right)}^{T} \end{array}\right)\left(\Delta \alpha (t),\Delta \gamma (t),\Delta \beta (t),\Delta \eta (t)\right).$$

To ensure the monitoring process is appropriately responsive without being overly sensitive, the MEWMA strategy is utilized, formulated as:(10)$$MEWMA\left(t\right)=\Lambda \Delta \bar{\theta }\left(t\right)+\left(1-\Lambda \right)MEWMA\left(t-1\right),$$
accompanied by a dynamic covariance matrix:$$V\left(t\right)=\Lambda^{2}Cov\left(\Delta \theta \left(t\right)\right)+{\left(1-\Lambda \right)}^{2}V\left(t-1\right),$$
where Λ is a smoothing or weighting matrix that moderates the impact of new observations on the cumulative average. Selecting lower values in the Λ leads to a more conservative approach, minimizing the influence of new data, whereas a higher value increases sensitivity to new observations. This adjustment helps manage false alarms, acknowledging the inherent variability in any process under observation. Commonly selected values in Λ range from 0.1 to 0.3.

Given that our focus is on detecting shifts in parameter values, we aim for a process mean of (0,0,0,0), which would indicate stability. In this multivariate context, we employ Hotelling’s $T^{2}$ statistic to identify significant shifts, defined as:(11)$$T^{2}\left(t\right)=MEWMA{\left(t\right)}^{T}V{\left(t\right)}^{-1}MEWMA\left(t\right).$$

With $n_{p}=1000$, the distribution of $T^{2}\left(t\right)$ approximates a $\chi^{2}$ distribution with four degrees of freedom. A $T^{2}\left(t\right)$ value exceeding the 0.95-quantile of 9.48 signifies a significant shift in the SEIRDV model, pointing to a notable shift in the SEIRDV process, indicating robust detection of changes in the process.

### 3.2. The MCUSUM Control Chart

In this section, we will discuss the application of the MCUSUM chart to monitor the parameters of the SEIRDV model described in Section 2. The MCUSUM control chart is an advanced statistical process control tool that effectively monitors the mean vector of a multivariate process over time. Originating from the Cumulative Sum (CUSUM) control chart introduced by Page in 1954 [26] for univariate data, the MCUSUM adaptation extends this concept to encompass multivariate processes [27]. This extension, known as the Multivariate CUSUM (MCUSUM) chart, facilitates the monitoring of changes in the mean vector, which consists of the means of each variable within the process. This capability is particularly valuable in settings requiring the simultaneous monitoring of multiple interrelated variables. In the context of epidemiology and disease modeling, it can help us identify significant changes in the model parameters, which may indicate shifts in the dynamics of the disease or the effects of interventions over time. Designed to detect significant shifts in the mean vector, the MCUSUM chart is particularly adept at identifying sudden and substantial changes within the process.

The MCUSUM control chart has several advantages for monitoring the SEIRDV model parameters during the COVID-19 pandemic. Firstly, the MCUSUM control chart is able to detect sudden and significant changes in the model parameters, which is useful for monitoring the impact of new variants and other unexpected events. Secondly, the MCUSUM control chart is able to account for the correlation between the different variables in the model, which is important for understanding the complex relationships between the model parameters. Finally, the MCUSUM control chart is a well-established statistical process control tool, which means that it is based on a solid foundation of statistical theory and practice.

Crosier (1988) proposed a MCUSUM control chart [28] where the statistic is given by:(12)$$MCUSUM_{i}=\left\{\begin{array}{cc} 0 & \mathrm{if}\;d_{i}\leq k\\ \left(MCUSUM_{i-1}+Y_{i}-\mu_{0}\right)\left(1-\frac{k}{d_{i}}\right) & \mathrm{if}\;d_{i}>k, \end{array}\right.$$
where $MCUSUM_{0}=0$, $d_{i}={\left[{\left(MCUSUM_{\left(i-1\right)}+Y_{i}-\mu_{0}\right)}^{\prime }\Sigma^{-1}\left(MCUSUM_{\left(i-1\right)}+Y_{i}-\mu_{0}\right)\right]}^{\left(1⁄2\right)}$, Σ is the variance-covariance matrix of *Y*, and *k* is a predetermined constant. $Y_{i}$ will be defined as previously, $\Delta \alpha {\left(t\right)}_{i}=\alpha {\left(t\right)}_{i}-\alpha {\left(t-1\right)}_{i}$, $\Delta \gamma {\left(t\right)}_{i}=\gamma {\left(t\right)}_{i}-\gamma {\left(t-1\right)}_{i}$, $\Delta \beta {\left(t\right)}_{i}=\beta {\left(t\right)}_{i}-\beta {\left(t-1\right)}_{i}$, and $\Delta \eta {\left(t\right)}_{i}=\eta {\left(t\right)}_{i}-\eta {\left(t-1\right)}_{i}$. These form a vector $\Delta \theta \left(t\right)={\left(\Delta \alpha {\left(t\right)}_{i},\;\Delta \gamma {\left(t\right)}_{i},\;\Delta \beta {\left(t\right)}_{i},\;\Delta \eta {\left(t\right)}_{i}\right)}^{T}$ with mean
$$\Delta \bar{\theta }\left(t\right)=\frac{1}{n_{p}}1_{n_{p}}^{T}\left(\Delta \alpha (t),\Delta \gamma (t),\Delta \beta (t),\Delta \eta (t)\right)$$
and variance
$$Cov\left(\Delta \theta \left(t\right)\right)=\frac{1}{n_{p}-1}\left(\begin{array}{c} \Delta \alpha {\left(t\right)}^{T}\\ \Delta \gamma {\left(t\right)}^{T}\\ \Delta \beta {\left(t\right)}^{T}\\ \Delta \eta {\left(t\right)}^{T} \end{array}\right)\left(\Delta \alpha (t),\Delta \gamma (t),\Delta \beta (t),\Delta \eta (t)\right).$$

The MCUSUM signals if any of $d_{i}$ exceeds the Upper Control Limit (UCL), where UCL is selected to achieve the desired in-control ARL.

The SEIRDV model parameters, including α(t), β(t), γ(t), and η(t), are considered as time-varying and are essential for understanding the spread and impact of COVID-19. Monitoring these parameters can provide valuable insights into the effectiveness of public health measures, vaccination campaigns, and other interventions.

To implement the MCUSUM chart for the SEIRDV model, we follow these steps:Initialize the MCUSUM control chart with an initial estimate of the parameters based on historical data. This estimate can be obtained using prior information or initial modeling.Continuously collect data on the number of infected individuals (I(t)), recovered individuals (R(t)), deaths (D(t)), and vaccinated individuals (V(t)) at each time point, *t*. These observed data points are used to update the model parameters and the MCUSUM chart.Employ a Bayesian methodology, such as the sequential sampling with particle augmentation technique, to estimate the time-varying parameters α(t), β(t), γ(t), and η(t) at each time step. This technique updates the posterior distribution of the parameters based on the observed data, providing a more accurate estimate of the parameters over time.Calculate the MCUSUM statistics for each parameter to monitor deviations from the expected values.Set control limits for each parameter, tailored to the desired levels of sensitivity and specificity. These limits are critical for determining when a parameter’s change is statistically significant and warrants further attention.Continuously monitor the MCUSUM statistics for each parameter. If any of the statistics exceed the defined control limits, it indicates a potential shift or change in the model parameters. At this point, further investigation and action may be required to understand the underlying causes of the change.As new data become available, continuously update the parameter estimates and the MCUSUM chart. This dynamic adjustment allows for real-time monitoring of changes in the pandemic’s dynamics or the effectiveness of health interventions, facilitating timely decision-making.

At the end, one can see that the MCUSUM chart provides a systematic approach to monitor the time-varying parameters of the SEIRDV model, enabling us to detect and respond to significant changes in the epidemiological dynamics of COVID-19. This monitoring process is crucial for informed decision-making in public health and policy interventions during the ongoing pandemic.

## 4. Simulation Study

To assess the algorithm’s effectiveness in detecting changes in model parameters, a simulation study was conducted. The simulation focused on the infection rate parameter, α, as it plays a critical role in monitoring pandemic dynamics. The study aimed to evaluate how different magnitudes of changes in α affect the algorithm’s ability to alert decision-makers to significant shifts. Both MEWMA and MCUSUM control charts were applied to monitor these changes. Precise parameter estimation in the SEIRDV model is crucial due to the complex and dynamic nature of disease spread, particularly with the added complexity of incorporating vaccination status. As new variants emerge and vaccination rates fluctuate, monitoring real-time changes in infection dynamics becomes more challenging. Pandemic data are inherently high-dimensional, involving multiple interrelated parameters that describe disease progression across different compartments, such as susceptibility, exposure, infection, recovery, death, and vaccination. This high-dimensionality makes it difficult to accurately estimate model parameters in real-time, which is critical for informing timely and effective public health interventions.

The augmented particle Markov Chain Monte Carlo (MCMC) scheme offers a solution by enabling precise, real-time parameter estimation. Unlike traditional methods, this approach efficiently samples from high-dimensional distributions, dynamically updating model parameters as new data become available. By integrating particle filtering techniques with MCMC, this scheme enhances the study’s ability to effectively monitor pandemic dynamics. It handles complex model structures and high-dimensional data more robustly, ensuring accurate detection of changes in pandemic progression. This accuracy is pivotal for making timely decisions regarding disease control measures, policy adjustments, and resource allocation. In essence, the augmented particle MCMC scheme significantly improves our capability to understand and respond to pandemic trends, providing critical insights for decision-makers tasked with navigating public health crises.

The simulated datasets are for a pandemic of 100 days with the event occurring on day 30. All simulated datasets are generated from the same initial values: S(0) = 9800, E(0)=10, I(0)=10, R(0)=0, D(0)=0, and V(0)=0. The parameter values are the same for all simulated datasets for the first 30 days, at which point α=0.00004 will change, and in our case specifically, become worse. The other parameters are defined as β=0.1, γ=0.15, η=0.5, ϕ=0, and ρ=0. The parameter α(t) is redefined at each time step based on the posterior distribution, which is updated dynamically using the most recent observed data. This ensures that the parameter reflects real-time changes in transmission dynamics. At day 30, α will be changed via a multiplier value to increase the infection rate. The following multipliers are considered: m=1,2,3,5,6, and 11. At day 30, $\alpha_{30+}=\alpha \times m$ and remains this value for the remainder of the pandemic time frame. For all simulations, the MCUSUM parameter *k* is set to 1. For each of the multipliers, 100 datasets were generated (using the same seeds for random number generation to ensure the first 30 days of data were identical) and the method was applied to each one, and two metrics were evaluated. The goal was to assess the method’s ability to detect changes, measuring the detection within one day (day 31) and within five days (days 31–35). The probability of detection across the 100 samples was computed for each time range.

Figure 1 shows example simulated datasets, with Panel (a) corresponding to no change in infection rate and Panel (b) corresponding to the case where the infection rate increases by a factor of m=5 in infection rate at day 30. Notice the dramatic change between the two simulated datasets, especially after day 30. The number of susceptible, *S*, drops considerably and the exposed and infected increase quickly. The monitoring method proposed does not see the *S* and *E* states as they are latent. Hence, we should not expect this method to signal immediately as one first becomes exposed before becoming infected.

Figure 2 shows the results of the simulation study. Panel (a) shows the power, the probability to detect a change at day 31 (first day since change) across multipliers and for MCUSUM and MEWMA with weights (W=0.2,0.3,0.4,0.5). Each line represents a different methodology/weight scheme. Notice that the MCUSUM technique dominates all of the MEWMA methods, meaning it has a much stronger ability to quickly detect a change in the infection rate. While it is better than the other techniques, it still has a low power rate at below 0.25 for a 3-times increase in infection rate and just barely a power of 0.5 for an 11-times (extreme) change in the infection rate. While higher power for the MCUSUM may be desirable, notice that at a multiplier of one, the curve is not below the Type I error rate of 0.05; hence, this method is far more likely to signal “false alarms”. Panel (b) shows the power, the probability to detect a change within the first five days of the change at days 31, 32, 33, 34, and 35 across multipliers and for both MCUSUM and MEWMA with the same weights as above. Again, the MCUSUM dominates the MEWMA method across all multiplier values, and it still takes a five-times increase in infection rate before the power is above 0.8. Similar to the first day detection, the MCUSUM approach does not preserve the Type I error rate of 0.05 and will likely produce more “false alarms”.

## 5. Real Application: State of Qatar

Datasets on the daily total of confirmed infection cases, fatalities, and recoveries for every nation are kept up to date by the World Health Organization, Johns Hopkins University, and other organizations. We use the suggested methodology to examine how the pandemic has changed in the State of Qatar. Johns Hopkins University provided all of the data for Qatar, and it is all freely available through the Johns Hopkins COVID-19 GitHub repository [29]. Starting on 22 January 2020, the GitHub site includes the daily cumulative number of infections that have been confirmed, the cumulative number of recoveries, and the cumulative number of deaths. The objective of the data analysis is to demonstrate and assess the suggested modeling technique and its use in monitoring the pandemic, since the variable coefficient approach allows the model parameters to rapidly adjust to changes in the data creation process. In contrast to the Infected state, which undergoes transitions from the Exposed state to the Recovered and Death states, the Recovered and Death states in model (1) do not exhibit any outward transitions and are cumulative in nature. Hence, the aggregated data for verified infections includes data for both the Recovered and Death conditions. Therefore, the quantity of individuals who are infected at a given time *t* may be expressed as I(t)=CI(t)−R(t)−D(t), where CI(t) represents the number of confirmed infections at time *t*. In the subsequent analysis, the variable that has been obtained will be denoted as “Active Infections” to distinguish it from the cumulative Infected value shown in the dataset. Figure 3 displays the daily statistics on Active Infections, Recovered, and Deaths for the State of Qatar from 29 February 2020. As a result of heightened testing, the quantity of Active Infections experiences a substantial rise on the twelfth day after a notably low initial level. The quantity of Active Infections thereafter reaches a steady state until day 30, at which point there is a rapid and exponential rise. Following a period of 90 days, there is a noticeable decrease in the quantity of Active Infections. The observed parallels in the patterns of the Recovered and the Deaths may be attributed to the temporal occurrence of infection preceding the subsequent recovery or demise. Until about day 90, both graphs show a marginal rise in the quantity of individuals who have recovered or have passed away. Subsequently, a consistent upward trend is visible. Also, this figure illustrates the cumulative impact of vaccination efforts on reducing susceptibility and aiding pandemic control.

### 5.1. A Comprehensive Analysis of Qatar’s Response to the Pandemic

The COVID-19 pandemic has had a significant impact on many nations, including the State of Qatar. With a total of 148,258 confirmed cases as of 26 January 2021, Qatar has become one of the most contaminated Middle Eastern nations since its first infected case on 29 February 2020. The fact that there have been relatively few fatalities in Qatar (248 instances) compared to the number of infections is a testament to the quality of the nation’s healthcare system. Based on a national risk assessment that took into consideration the results of the World Health Organization’s global risk assessment and prioritized strengthening capacity to decrease or eliminate COVID-19 health concerns, Qatar developed an outstanding, adaptable strategy for risk management. Its well-organized healthcare system aided the country’s rapid response to the worldwide epidemic. Among the various preventative measures put in place by the government at the outset of the epidemic was stricter border control, in an effort to identify cases as soon as they emerged. Some of the measures taken included the installation of thermal screening machines at seaports and Hamad International Airport beginning in January 2020, with the first quarantine facilities operating on 1 February of the same year [30].

On day ten, 9 March 2020, Qatar banned travel from fifteen nations: Bangladesh, China, Egypt, India, Iran, Iraq, Iraq, Italy, Lebanon, Nepal, Pakistan, the Philippines, South Korea, Sri Lanka, Syria, and Thailand. The country also closed all schools and institutions. In an expansion of its travel restrictions, Qatar added three more countries: France, Spain, and Germany on 14 March 2020 (day 15) [31,32]. On 21 March 2020, in an effort to contain the coronavirus, the Ministry of Municipal and Environmental Control shut down all public parks and beaches. The decision to temporarily shut all public restaurants, cafés, food outlets, and food trucks was made by the Ministry of Commerce and Industry (MoCI) on 23 March 2020 (day 24). On day 28, 27 March 2020, the MoCI also decided to shut down any non-essential enterprises [31,32].

The Ministry of Public Health (MoPH) announced on 8 April, 2020 (day 40) that two health centers, one in Umm-Salal and one in Gharrafat Al-Rayyan, would be designated by the Primary Health Care Cooperation to screen, test, and quarantine COVID-19 patients as the number of infected cases continued to rise. A hotline for psychiatric assistance was also launched by the MoPH on 9 April 2020 (day 41). On day 95, 3 June 2020, four persons and their families were permitted inside a car, and private sector workers had their working hours extended from 7:00 a.m. to 8:00 p.m. Additionally, it was reported in early June 2020 that the nation would begin its partial reopening on 15 June 2020 (day 108) and the second part on 1 July 2020 (day 123).

While establishing a real-time monitoring system of the infection, recovery, and mortality rates, it is important to take into account these proactive measures performed by the government, as they alter the pandemic dynamics. To help stakeholders in Qatar keep an eye on the COVID-19 pandemic, we provide the suggested model in the following section as a data-driven forecasting tool.

### 5.2. Results of the Analysis

It is common practice to seek prior distributions that are either diffuse or minimally informative when attempting to define them. Since the data do not cover all model transitions in the first time periods, adequately informative distributions need to be specified for the SEIRDV model. For instance, at the beginning of the study, neither recoveries nor deaths are recorded. Consequently, the parameters that control the transitions from the infected to the recovered state and from the infected to the death state cannot be determined by data from these states at this point in time. Another thing to think about is that if the parameter values are not picked well, dynamical systems may reach a stable point extremely quickly, such as when the sickness stops spreading or when everyone dies. To make sure the model’s initialization phase goes well, we tuned it to the time periods up to the first death, and we then picked the prior distribution parameters. As a consequence, the following specification was obtained: α(t)∼Exp(2/4,450,000). With respect to time, β(t)∼Exp(1/105), γ(t)∼Exp(1/14), and η(t)∼Exp(1/9500). The values that [10] selected are exact replicas of these.

The following parameters were used to run the SEIRDV model in Qatar: In the initial state, S(0) = 2,782,000, with E(0)=3, I(0)=1, R(0)=0, and D(0)=0. This study used MATLAB R2020a to code the SEIRDV model and conduct the sampling method described in [10]. The PC used has an Intel Core i7-7700 CPU running at 3.60 GHz and 8 GB of RAM. With $n_{c}$ =10,000, $n_{p}$ = 1000, and $n_{b}=10$, the sampler was executed at each time step. Thus, $n_{p}$ candidate particles were produced at the start of the next time step from $n_{b}$ samples generated in the vicinity of each $n_{p}$ sample; this could be achieved, for instance, by randomly sampling from a small-variance Gaussian distribution centered at each $n_{p}$ sample. For T=135 time steps, the model’s calculation duration is around 60 min. Keep in mind that, because of the population size, there are far less people in each model compartment. Because of this, many of the calculations are done more quickly, particularly when working with large factorials related to the Poisson distribution.

The predictions of the coefficients α(t) (panel a), β(t) (panel b), γ(t) (panel c), and η(t) (panel d) for the Qatar dataset are shown in Figure 4. The period from day 90 to day 95 is significant since it is at this time that the active infections I(t) showed a significant decrease (see to Figure 3). The very condensed distribution of α(t) in this time frame is shown by the tiny credible intervals, which demand attention. Both β(t) and η(t) show this in further detail. Looking at γ(t) throughout this time period also reveals a significant increase in the recovery rate, with very small credible intervals after day 80. Another increase occurs around day 115, and the recovery rate continues to climb sharply thereafter.

With 95% posterior prediction limits for Active Infections (panel a), Recovered (panel b), and Deaths (panel c), the model fitted to the data is shown in Figure 5. At first glance, it would seem that all three models provide a very good match to the data. Note in particular that, in comparison to Figure 4, there is a significant decrease in active infections around day 90.

To evaluate the model’s accuracy, a Pseudo-$R^{2}$ metric was computed using the formula [33]:$$Pseudo-R^{2}=1-\frac{\sum_{t=1}^{n}{\left[I\left(t\right)-\hat{I}\left(t\right)\right]}^{2}+\sum_{t=1}^{n}{\left[R\left(t\right)-\hat{R}\left(t\right)\right]}^{2}+\sum_{t=1}^{n}{\left[D\left(t\right)-\hat{D}\left(t\right)\right]}^{2}}{\sum_{t=1}^{n}{\left[I\left(t\right)-\bar{I(t)}\right]}^{2}+\sum_{t=1}^{n}{\left[R\left(t\right)-\bar{R(t)}\right]}^{2}+\sum_{t=1}^{n}{\left[D\left(t\right)-\bar{D(t)}\right]}^{2}},$$
where $\bar{I}$, $\bar{R}$, and $\bar{D}$ represent the mean values of *I*, *R*, and *D* over time, respectively, and $\hat{I}\left(t\right)$, $\hat{R}\left(t\right)$, and $\hat{D}\left(t\right)$ are the medians of the posterior predictive distributions for *I*, *R*, and *D* at each time point (and hence are functions of time).

For assessing the model’s uncertainty, the percentage of data points falling within the predictive intervals was calculated as:$${\hat{P}}_{\mathrm{fit}}=\frac{\sum_{t=1}^{n}I_{\left\{I\left(t\right)\in {\hat{C}}_{I(t)}\right\}}+\sum_{t=1}^{n}I_{\left\{R\left(t\right)\in {\hat{C}}_{R(t)}\right\}}+\sum_{t=1}^{n}I_{\left\{D\left(t\right)\in {\hat{C}}_{D(t)}\right\}}}{3n}$$
where $\hat{C}I(t)$, $\hat{C}R(t)$, and $\hat{C}D(t)$ are the 95% predictive intervals for I(t), R(t), and D(t), respectively, and IA is an indicator function that equals one if the condition A is met.

The Pseudo-$R^{2}$ value of 0.9999 indicates an exceptional correlation between the observed data and the median predictions. The proportion of observations within the 95% predictive bands, ${\hat{P}}_{\mathrm{fit}}=0.8394$, suggests the model captures the majority of data points, although it indicates a lower level of uncertainty than expected. Nonetheless, this proportion is still considerable, with approximately 84% of observations enclosed by the predictive intervals.

The MCUSUM statistic $d_{i}$, plotted over time for the Qatar dataset, shows stability up to day 40, as indicated by $T^{2}\left(t\right)$ values remaining below the control limit. Subsequent days, including 40, 44, 47, 64–69, 71, 77, 95, and 123, are marked by significant changes, corresponding to alterations in the infection and death rates, as evidenced in early data points and specific intervals of high volatility. Notably, spikes in the infection rate and notable increases in the recovery rate are observed, reflecting the dynamic evolution of the pandemic in response to policy changes and public health measures in Qatar.

These findings align with the trajectory of the COVID-19 pandemic in Qatar, highlighting key moments, such as the establishment of healthcare facilities on day 40, the dramatic increase in daily cases in early May, the relaxation of restrictions on day 95, and the commencement of the second phase reopening on day 123. These milestones are indicative of the pandemic’s evolving nature, as captured by our model’s analytical insights.

To examine how the sensitivity to the smoothing parameter (Λ) in the MEWMA chart affects the detection of signals, additional analyses were conducted using Λ values of 0.1, 0.15, 0.25, and 0.3, which are standard selections for this parameter. With Λ set at 0.1, the detection algorithm identified changes on days 44, 47, 69, 77, and 123. Given that a lower Λ emphasizes the prior mean over recent observations, fewer days being flagged as significant is anticipated. For Λ at 0.15, changes were noted on days 40, 44, 47, 64, 68, 69, 71, 77, and 123, as in Figure 6. It is notable that day 95 was not detected as a point of change for both Λ=0.1 and Λ=0.15, despite evident shifts in the process across nearly all charts, suggesting that these settings may be overly cautious. With Λ=0.25, the days highlighted were 40, 44, 47, 64, 65, 67, 68, 69, 71, 77, 95, and 123, mirroring the outcomes when Λ was at 0.2. At Λ=0.3, the algorithm flagged days 40, 44, 47, 64, 65, 67, 68, 69, 71, 72, 74, 77, 80, 95, and 123, with days 72, 74, and 80 newly identified, indicating increased volatility in α(t) during this period. Generally, opting for less conservative Λ values yields logical days for signaling. In the context of a pandemic, a monitoring system with higher sensitivity could be advantageous for policymakers by alerting them to the efficacy of interventions sooner.

Figure 7 represents the signals for the SEIRDV model parameters (α,β,γ,η and ϕ) over a period of 500 days. Each parameter is plotted against time, showing the deviations from their expected values. From this graph, one can see the following:Initial Period (Days 0–100): The parameter α (red line) shows significant fluctuations, indicating changes in the transmission rate. This aligns with the initial phase of the pandemic where interventions and public behavior may have varied significantly. Parameters (β,γ,η and ϕ) exhibit minor fluctuations, suggesting relatively stable infection, recovery, and death rates initially.Middle Period (Days 100–400): During this period, the parameters generally remain close to zero, indicating a period of relative stability in the model parameters. This may correspond to a phase where the public health measures were effective and the spread of the virus was relatively controlled. Minor fluctuations in the signals of (α,β,γ,η and ϕ) suggest occasional changes in transmission, infection, and death rates, possibly due to minor policy changes or the emergence of new variants.Final Period (Days 400–500): An increase in the signal fluctuations for (α) is observed again, indicating another phase of significant changes in the transmission rate. This could correspond to the emergence of new variants, changes in public behavior, or modifications in public health policies. Fluctuations in other parameters (β,γ,η and ϕ) become more noticeable towards the end, indicating increased variability in the infection, recovery, death, and vaccination rates.

Therefore, this Figure effectively highlights periods of significant changes in the pandemic’s dynamics as captured by the SEIRDV model parameters. The initial and final periods show more variability, likely due to the initial outbreak and subsequent waves of the pandemic. The middle period of relative stability might indicate effective control measures. This visual analysis supports the findings discussed in the text and underscores the importance of continuous monitoring to detect and respond to significant changes in the pandemic dynamics. These findings align with the trajectory of the COVID-19 pandemic in Qatar, highlighting key moments such as the establishment of healthcare facilities on day 40, the dramatic increase in daily cases in early May, the relaxation of restrictions on day 95, and the commencement of the second phase reopening on day 123. These milestones are indicative of the pandemic’s evolving nature, as captured by our model’s analytical insights.

## 6. Discussion and Conclusions

This study utilized MEWMA and MCUSUM control charts to effectively monitor SEIRDV model parameters during the COVID-19 pandemic in Qatar. These control charts proved to be practical tools for detecting changes in model parameters over time, thereby aiding decision-making related to public health interventions and measures. The dynamic treatment of α(t) as a time-varying parameter demonstrates the flexibility and adaptability of the SEIRDV model to real-world scenarios. Future studies may explore alternative functional forms or include additional covariates to further refine this parameter.

However, it is essential to acknowledge the limitations of the SEIRDV model used in this study. The assumption of a homogeneous population, where all individuals share equal susceptibility, may not fully capture real-world variations across different demographic groups. Future research should incorporate demographic and other relevant factors to enhance the model’s accuracy and predictive power. Another limitation lies in the assumption of constant movement rates between compartments. In reality, factors such as testing availability and healthcare resources can affect these transitions, impacting the model’s predictions. Future studies should consider more realistic assumptions to improve accuracy. Despite these limitations, the study underscores the potential utility of MEWMA and MCUSUM control charts in monitoring SEIRDV model parameters. These methods effectively signal changes, enabling public health officials to make timely and informed decisions regarding interventions and measures.

This study also provides a comparative analysis of MCUSUM and MEWMA methods. While the MCUSUM method excels in detecting significant shifts in multivariate settings, it lacks the intuitive simplicity of MEWMA, especially in single-variable scenarios. MEWMA’s resemblance to its univariate form enhances its intuitiveness and ease of implementation, making it particularly advantageous in applications requiring flexibility across varying numbers of variables or where user-friendliness is critical. While MCUSUM is highly sensitive to abrupt and significant changes across multiple variables, MEWMA offers greater versatility and accessibility, particularly for users with limited statistical expertise. Therefore, the choice between these methods should be guided by the complexity of the data and the user’s familiarity with multivariate techniques.

This research also emphasizes the importance of the continued monitoring and analysis of COVID-19, both in Qatar and globally. Such monitoring is critical for effective public health responses and managing ongoing pandemics.

Finally, this study highlights that statistical process control techniques, as applied to epidemiological models, provide valuable insights into disease dynamics and intervention effectiveness. Although the focus was on COVID-19, these techniques can be adapted to monitor other infectious diseases, offering a proactive approach to public health management. This has the potential to save lives and mitigate the impact of future outbreaks.

## Figures and Tables

**Figure 1 ijerph-21-01580-f001:**
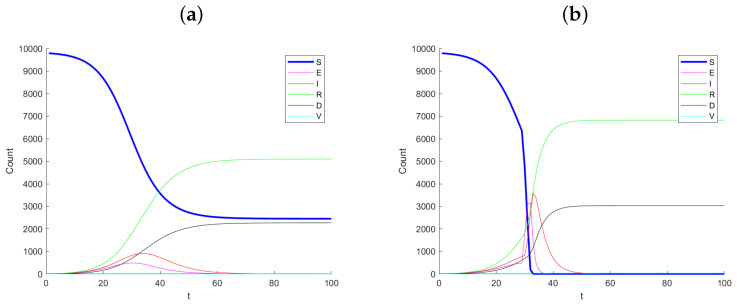
Example simulation paths for (**a**) no change (control) and (**b**) a 5-times change in infection rate at day 30.

**Figure 2 ijerph-21-01580-f002:**
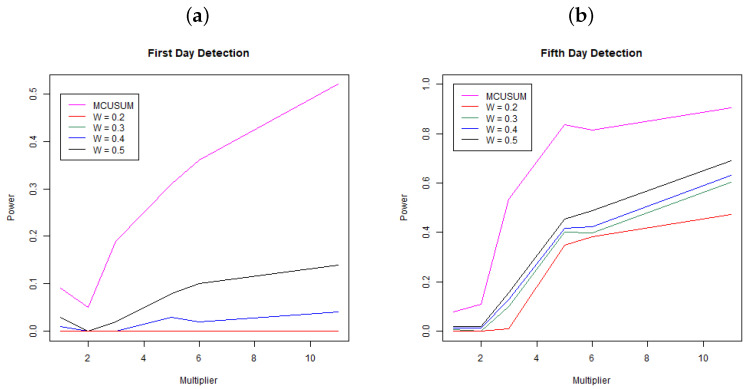
Probability of detection for MEWMA (W = 0.2, 0.3, 0.4, 0.5) and MCUSUM. Panel (**a**) shows the first day detection and Panel (**b**) shows the detection rate within five days of infection. Each point is the mean of the simulation results based on 100 simulated datasets at each simulation setting.

**Figure 3 ijerph-21-01580-f003:**
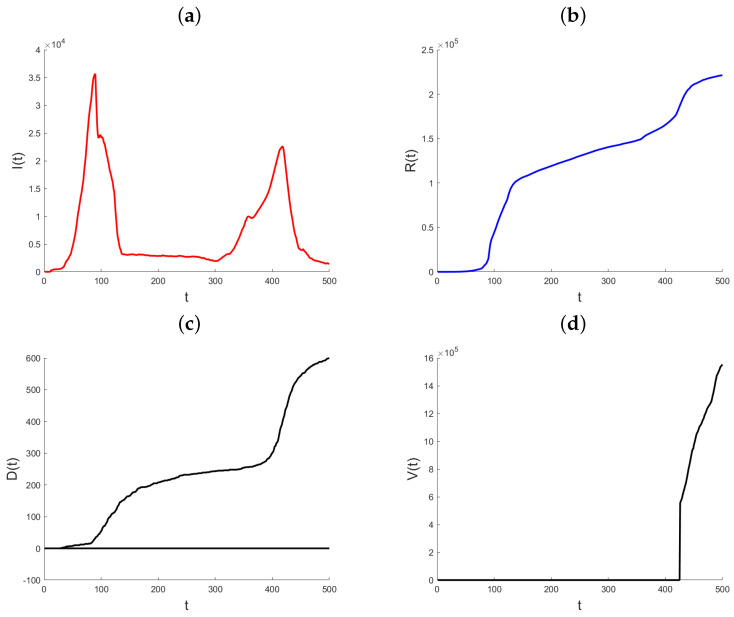
Plot of Active Infections (**a**), Recovered (**b**), Deaths (**c**), and Vaccinated (**d**) for the State of Qatar data. *t* represents the number of days since the initial outbreak.

**Figure 4 ijerph-21-01580-f004:**
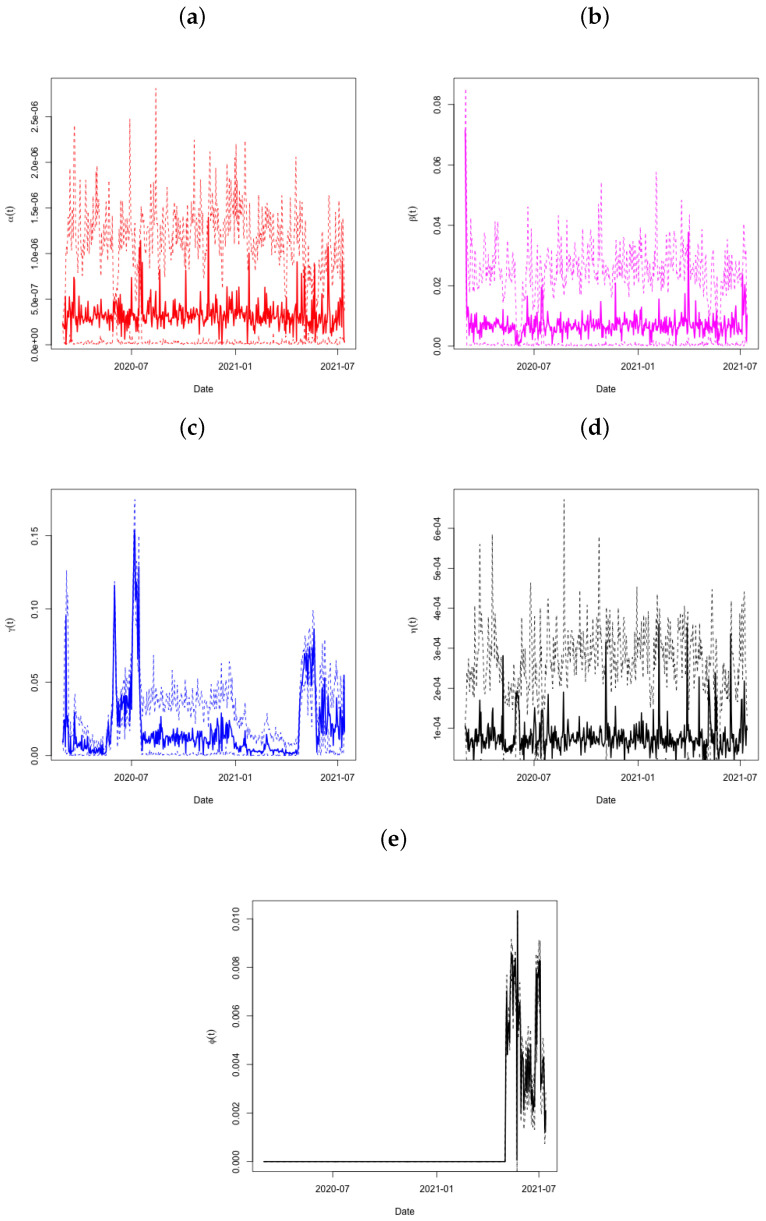
Plots of α(t) in Panel (**a**), β(t) in Panel (**b**), γ(t) in Panel (**c**), η(t) in Panel (**d**), and ϕ(t) in Panel (**e**) across time, with associated 95% credible intervals for the State of Qatar data.

**Figure 5 ijerph-21-01580-f005:**
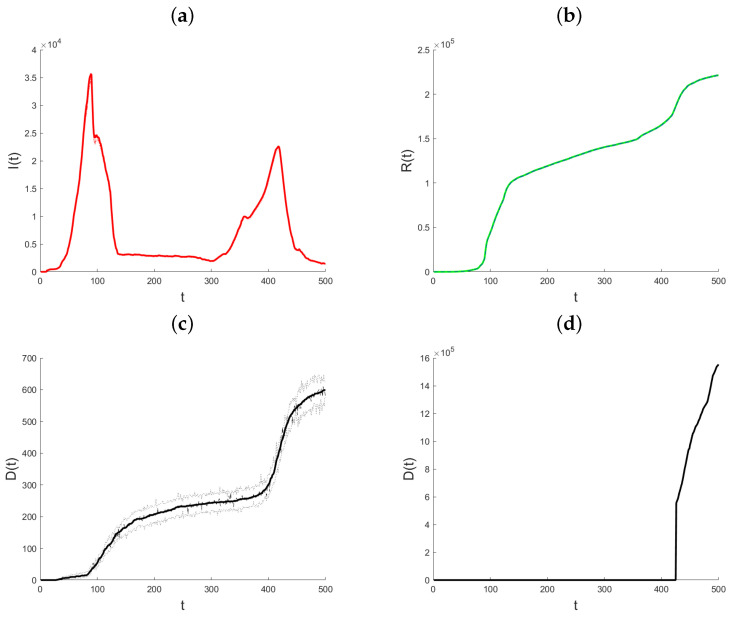
Plots of the data-fitted model with 95% posterior prediction bounds for Active Infections (**a**), Recovered (**b**), Deaths (**c**), and Vaccinated (**d**) for the State of Qatar data.

**Figure 6 ijerph-21-01580-f006:**
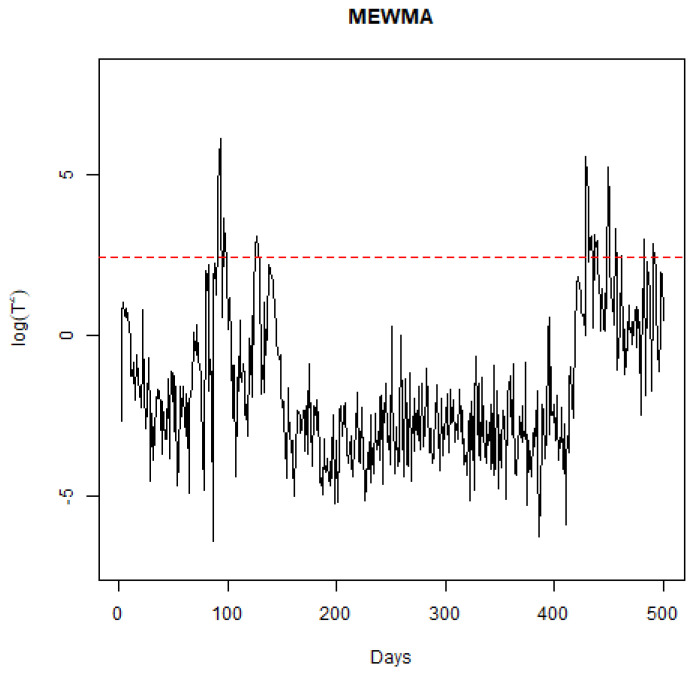
Plot of the MEWMA $log(T^{2})$ statistic through time.

**Figure 7 ijerph-21-01580-f007:**
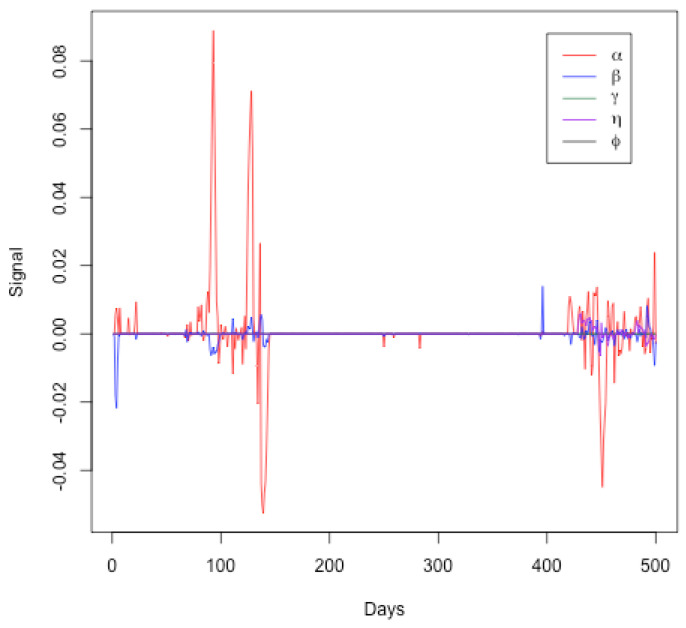
Signals for SEIRDV model parameters (α,β,γ,η and ϕ) over a period of 500 days.

## Data Availability

Data are available from the authors upon request.

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
