# Peer review of "Multivariate Techniques for Monitoring Susceptible, Exposed, Infected, Recovered, Death, and Vaccination Model Parameters for the COVID-19 Pandemic for Qatar"

_ijerph, 2024, doi:10.3390/ijerph21121580_

Round 1
Reviewer 1 Report
Comments and Suggestions for Authors
Review of manuscript:
"Multivariate Techniques for Monitoring SEIRDV Model Parameters for the COVID-19 Pandemic for Qatar."
The paper combines two methodologies, the Multivariate Exponentially Weighted Moving Average (MEWMA) and the Multivariate Cumulative Sum (MCUSUM), for pandemic monitoring. The monitoring is done via the dynamic (temporal variation of) model parameters of the SEIRDV model, with application to CIVID-19 data from the State of Qatar, where significant shifts are highlighted.
The model parameters are updated using MCMC sampling algorithm with particle augmentation. Real data analysis is provided with the COVID-19 count over time in the State of Qatar.
The paper is valuable and adds value to the literature. However, there are few comments that must be addressed.
Below are my comments, organized into two major and minor categories.
Major comments:
I am looking at the part of the manuscript where the augmented particle MCMC is described in-depth. However, that is not apparent. Please describe the "Bayesian approach" thoroughly as mentioned in the "Introduction" section, on page 3, line 114.
Give reference for Equations 1-6. Equation 7 seems to be a novelty. Please explain the idea and the meaning of \rho.
Page 6: last line: in the equation the meaning of n_p is not described. Later on, the indices n_c and n_b are shared. Please describe all those values and their meaning. Also include the \bold{1}_{n_p}. The authors should introduce the parameter vector \theta(t), that way, the \bar{theta}_t would be easily accessible. Explain the weighting in the EWMA.
Page 7: equation 10: the \Lambda smoothing parameter is a vector or matrix. Please revise. It is looked like that portion is borrowed from the univariate EWMA description, with values between 0.1 to 0.3.
Page 8: line 310: remove the period, and adjust the sentence.
Page 8: line 312: introduce the \Delta \theta (t). That will make the reading flow more concordant.
In the end, the MEWMA seems to outperform the MCUSUM in all cases. Is this a new finding? Has literature provided evidence for such result? Please give a clear explanation.
Authors should mention that the simulations and real life examples are built under the assumption that the MCMC model is homogeneous, or in the time-varying in control process. What would be the model estimates and effects for out of control mean values? Would the monitoring work?
Authors should try to describe Figures 3 and 4 in the same x-axis: in number of days or in calendar dates.
Page 13: lines 496-497: explain how ref [10] uses those values of \alpha, \beta and so on.
Page 13: line 522: the value of \rho was not mentioned. Please specify. If not need, then what was the reason in adding the vaccination in equation 7.
Minor comments:
Page 1: lines 21-23: use the past tense.
Page 1: line 34: add a reference after the word "symptoms."
Page2: line 54: spell out SEIRDV, before using the acronym. Same comment for MEWMA and MCUSUM. On line 59, the authors could introduce the acronym "CCs" and then on lines 62-63, use the acronym itself.
Page 2: lines 73-74: the sentence needs rewriting.
Page 4: line 179: please rewrite the "\lambda_{S}(I)." It looks like something else is meant. Also be consistent with the lower and upper "S" since the index "s" in line 177 looks like a lower case format.
Page 5: equation 8: the "\lambda_{I}" should have the "t" added in. Please end with a period, or complete the sentence after the comma. Also, mention something about the "\lambda_{V}(t)."
Page 6: line 245: authors should give the reference for the "Roberts" 'paper.
Page 6: line 258: end the sentence with a period instead of column.
Page 7: line 288: give reference with the "Page."
Page 8: Equation 12 must end with a comma.
Page 8: line 309: write the "k" with a math symbol (same in page 9 line 385) and specify if it is positive. Specify that value in the simulations setups.
Page 8: line 314: remove the period, and adjust sentence.
Page 9: lines 380: give the alpha value "0.00004" in that line, not in line 381.
Page 9: line 392: please specify that "m=5" is what is meant with Panel (b).
Page 11: last line before the subsection 5.1. Mention something on the vaccination plot described in Figure 3. Talking about Fig 3, in the label of Fig 3, please add a comma between "(b)" and "Deaths."
Page 12: line 469: I think "non essential" is more appropriate than "useless."
Page 13: add a commas after the equations after line 523 and line 528. Also give references if these equations are well known, or share the derivations.
Page 16: line 564: the symbol \phi is difficult to relate with what's been written before. Please give details.
Pages 18-19: references 4, 5 22, 24 and 29 were not found. Please explain.
Reviewer 2 Report
Comments and Suggestions for Authors
The paper conducts an analysis of the implementation and accuracy of the extended SEIR model, which includes vaccination and mortality, using two methodologies: the multivariate exponentially weighted moving average (MEWMA) control chart and the Multivariate Cumulative Sum (MCUSUM). My overall impression is that this is a good, well-written piece of work. However, I have some issues to present below. The order of the queries does not imply priority:
• The keywords should also include terms related to SEIR models.
• In the introduction, the detailed narrative of COVID-19 spread around the world and the virus characteristics may not be necessary. Instead, a more detailed overview of SEIR models used in COVID analysis and/or statistical techniques used to calibrate SEIR parameters should be provided. A table listing relevant works, theoretical approach (SEIR type), and multivariate techniques (used for parameter adjustment) would be helpful.
• The citation style is inconsistent (for example, line 74 mixes APA style with that of IJERPH). Additionally, citations should start with [1] and then continue with [2], [3], etc. Thus, the first work should be Rezabakhsh et al. (2020).
• Equations (1)-(7) suggest that parameters alpha, beta, gamma, etc., are constants, but this is not the case later. I suggest making these parameters dependent on t from the beginning. To distinguish variables from parameters, consider including “t” as a subscript for parameters.
• The equations need review. For example, the equation on line 179 is incorrect. Between line 181 and equation (8), a linking phrase like “where” or “being” should be added. Also, the Poisson parameter dynamics (equation (8)) should be well explained (whether derived from equations (1)-(7) or referenced). Equation (8) and subsequent equations should also be numbered in the same style as equations (1) through (7).
• Although the English is generally correct, some expressions should be revised for issues, such as informal expressions like “aren’t.”
• In the numerical application to Qatar, clearly stating the start and end dates of the study and explicitly noting the number of people observed in each state at some significant dates would clarify the data presentation.
• The metrics used to measure the model’s fit should be referenced. Are they widely accepted metrics for measuring fit quality?
• Regarding the model, we know it describes well, ex post, the evolution of infections, deaths, recoveries, etc. However, the authors do not report any predictive capability. Also, it is unclear how the methodology improves SEIRDV model fitting compared to existing alternatives. This should be reported, for example, by estimating SEIRDV with alternative methodologies or implementing a simpler SEIR model and discussing the improvements their method provides.
• In line with the previous paragraph, the discussion should put the results in perspective with respect to SEIRDV fitting methodologies or SEIR models and highlight the improvements offered by their approach compared to existing ones.
Reviewer 3 Report
Comments and Suggestions for Authors
Please see the attachment.

Normal.
Round 2
Reviewer 2 Report
Comments and Suggestions for Authors
Some of the suggestions have not been taken into account. For example, either the parameter alpha is constant, or alpha(t) must be redefined.
Please make the evaluator's task easier by marking in the text any issues that require adding content. For those that involve reduction, specify exactly what changes have been made.
It is not clear what the contribution of this work is. Obviously, monitoring infections is not new, and it is unclear how this improves upon existing methods. Since the objective is to "monitor," it is also unclear what the authors mean by monitoring. Typically, monitoring involves making short-term predictions (e.g., a week ahead). With or without Bayesian techniques, infections have been "monitored" for many years. The authors still do not discuss what the presented work contributes to already established techniques.
Author Response
Comment#1: Some of the suggestions have not been taken into account. For example, either the parameter alpha is constant, or alpha(t) must be redefined.
Response
I appreciate your feedback and the opportunity to clarify the approach regarding the parameter in our research. The parameter , representing the daily transmission rate, plays a crucial role in modeling the dynamic nature of the SEIRDV framework. We recognize the concern about whether , is being treated as a constant or a time-varying parameter. In our study, , is explicitly modeled as a time-varying parameter to capture the evolving transmission dynamics influenced by external factors such as public health interventions, behavioral changes, and the emergence of new variants.
Comment#2: Please make the evaluator's task easier by marking in the text any issues that require adding content. For those that involve reduction, specify exactly what changes have been made.
Response
Thank you for your suggestion. All new changes have been bolded in the text to facilitate the evaluation process.
Comment#3: It is not clear what the contribution of this work is. Obviously, monitoring infections is not new, and it is unclear how this improves upon existing methods. Since the objective is to "monitor," it is also unclear what the authors mean by monitoring. Typically, monitoring involves making short-term predictions (e.g., a week ahead). With or without Bayesian techniques, infections have been "monitored" for many years. The authors still do not discuss what the presented work contributes to already established techniques.
Response
Thank you for this insightful comment. We appreciate the opportunity to clarify the contribution of our work and how it advances existing methodologies in pandemic monitoring. In this study, "monitoring" refers to real-time tracking and detection of significant changes in the parameters of the SEIRDV model, such as transmission (), recovery (), and death () rates. Unlike traditional monitoring approaches that focus solely on short-term predictions of infections, our approach provides continuous assessment of changes in epidemiological dynamics, allowing public health officials to detect and respond to shifts in pandemic dynamics (e.g., emergence of new variants, impact of interventions). Also, it enables proactive decision-making rather than reactive responses, especially during periods of volatility.
Hence, in this study, we combine Multivariate SPC tools (MEWMA, MCUSUM) and Bayesian estimation to monitor multiple dynamic parameters simultaneously. Unlike traditional methods focusing on static infection counts, this dual approach balances sensitivity (gradual shifts) and robustness (abrupt changes). Also, the dynamic Bayesian framework allows continuous updates of parameters, providing early warnings and enabling short-term predictions based on posterior distributions. In addition, the practical utility of application to Qatar's data demonstrates how this approach identifies parameter shifts tied to specific public health interventions, offering actionable insights.

Reviewer 3 Report
Comments and Suggestions for Authors
The author carefully revised the paper according to the comments of the reviewers.
Author Response
Thank you for your acknowledgement. We really appreciate all your efforts and time.